# Interaction between Shock Waves Travelling in the Same Direction



**Pavel Bulat** [1,*] , **Konstantin Volkov** [2] **and Igor Volobuev** [1]

1   Scientific Research Laboratory of Unmanned Aerospace Transport Systems, Baltic State Technical University, 190005 Saint Petersburg, Russia; volobuev_ig@mail.ru
2   Faculty of Science, Engineering and Computing, Kingston University, London SW15 3DW, UK; k.volkov@kingston.ac.uk
*   Correspondence: pavelbulat@mail.ru

**Abstract:** In this paper, we study the intersection (interaction) between several steady shocks traveling in the same direction. The interaction between overtaking shocks may be regular or irregular. In the case of regular reflection, the intersection of overtaking shocks leads to the formation of a resulting shock, contact discontinuity, and some reflected discontinuities. The type of discontinuity depends on the parameters of incoming shocks. At the irregular reflection, a Mach shock forms between incoming overtaking shocks. Reflected discontinuities come from the points of intersection of the Mach stem with the incoming shocks. We also consider the possible types of shockwave configurations that form both at regular and irregular interactions of several overtaking shocks. The regions of existence of overtaking shock waves with different types of reflected shock and the intensity of reflected shocks are defined. The results obtained in the study can potentially be useful for designing supersonic intakes and advanced jet engines.

**Keywords:** shock wave; contact discontinuity; interference





## 1. Introduction

In a stationary supersonic flow, two types of shock wave reflection are distinguished—regular (two-wave configuration) and Mach reflection (three-wave configuration) [1]. In a certain range of parameters, the existence of two possible solutions and the associated phenomenon of hysteresis are allowed. The location of shock waves depends on the freestream Mach number, the angle of incidence of the shock wave, and the effective adiabatic exponent. The most studied instance is the case of a reflection of an oblique shock wave from a flat wall, as well as the reflection of a plane shock wave from a wedge.

Oblique shock waves in one direction, as well as shock waves travelling in the same direction, turn the flow in one direction. For example, flow turns in counter-clockwise direction are shown in Figure 1a, where M is the Mach number, $T$ is the point of intersection of the shock waves, $R$ is the reflected discontinuity, $\sigma_1$ and $\sigma_2$ are the overtaking shock waves, $\sigma_3$ is the main shock, $\tau$ is the tangential discontinuity, $\leftarrow$ is the left discontinuity, and $\rightarrow$ is the right discontinuity. Figure 1b shows a one-dimensional unsteady analogue of the interference of oblique shock waves in one direction. Two normal shock waves move in one direction. Shock wave $D_1$ runs faster than $D_2$ because the velocity $u$ behind the wave $D_2$ is not equal to zero; therefore, this shock wave is referred to as overtaking. By analogy with one-dimensional shock waves, oblique shock waves in one direction are also often referred to as overtaking. A systematic analysis of the interference of shock waves was performed in [2].

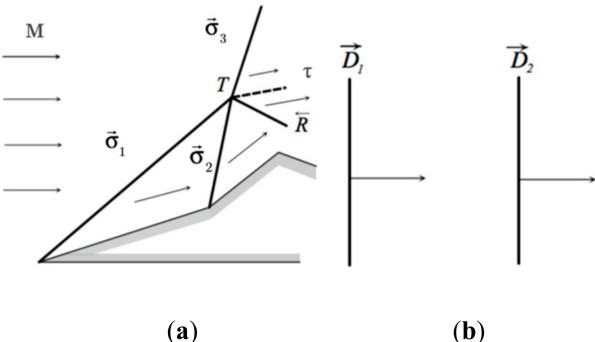

**Figure 1.** Interference of shock waves in one direction: (**a**) oblique shock waves in one direction, (**b**) overtaking one-dimensional shock waves $D_1$ and $D_2$ (arrows show the direction of motion of shock waves).

One triple point $T$ (Figure 1a) can receive many overtaking shock waves. With an increase in their number, such a configuration tends to a centered isentropic compression wave. This configuration is ideal for compressing flow in hypersonic air intakes.

The first published works on the study of overtaking shock waves appeared in the 1950s and 1960s, and were associated with the appearance of two- and three-shock adjustable air intakes of supersonic aircraft [3,4]. The design of supersonic aircrafts caused an intensification of the study into interactions between individual waves and their discontinuities, such as rarefaction waves with a shock wave, a shock wave with a contact discontinuity (refraction of a shock wave), etc. Shock wave structures can be combined into more complex systems consisting of triple configurations of shock waves, united by bridge-like shocks, the flow behind which is subsonic. A number of experimental and theoretical investigations were carried out to study the interaction of one-dimensional traveling waves and discontinuities. These studies include the investigation of refraction of a traveling shock wave at a contact discontinuity [5], the interaction between overtaking shock waves [6], the interaction between a shock wave and a rarefaction wave [7], and the refraction of a rarefaction wave [8].

One of the triple configurations (TK3) that corresponds to a case in which a reflected discontinuity is a characteristic is discussed in [9]. Ashock wave structure composed of overtaking shocks of the same intensity is of the lowest level in terms of total pressure loss [10]. Such a shock wave structure is optimal for supersonic air intake. Studies of the domains of existence in shock wave structures formed during the regular interference of overtaking shocks has been carried out in [1,11]. Attempts were made to study irregular shock wave structures. Subsequently, the theory of constructing optimal shock wave structures was developed in [12,13]. In particular, various criteria for the optimization of a shock wave structure, composed of overtaking shocks, were formulated [14,15]. Finally, this theory was presented in relation to the external aerodynamics of supersonic aircraft in [16]. The characteristic shock wave structure of overtaking shocks belongs to one of the triple configurations for shock waves [17,18].

In addition to supersonic air intakes, it is also necessary to mention problems related to external aerodynamics that are associated with overtaking shocks. These include the emergence of local supersonic regions with λ-shaped shock waves on the upper surface of the wing profile during a wave crisis, as well as the formation of overtaking shocks on a needle extended into a supersonic flow in front of the body in order to reduce resistance, or when supplying energy to a supersonic flow with the formation of plasma [19]. Irregular refractions of a shock wave on a mixing layer are considered in [20].

Supersonic flows are complex numerical simulation problems. The problem regarding the regular or Mach reflection of a shock wave from a wall is used in many studies to establish the accuracy of different schemes and the performance of numerical methods [21–25].

Theoretical issues related to the study of the reflection of a shock wave from a solid wall are presented in [1] on the basis of the shock polar method.

Regions of existence related to overtaking shock waves with different types of reflected shock, as well as the intensity of reflected shocks, have not been investigated. In many studies, only general relations are provided. Solutions to the problems of shock wave structures on the plane of a shock polar, the determination of the domains of existence, the intensity of a reflected discontinuity, and the use of shock wave structures in external compression air intake are all considered in the study.

## 2. Regular Refraction

An oblique shock wave in a supersonic flow is a gas-dynamic discontinuity (stationary shock wave) located at an angle to the incident flow. A shock located at an angle σ compresses the incoming flow and turns it on a certain angle β (Figure 2). Angle β is taken with a plus sign if it is plotted counter-clockwise from the free stream velocity vector (such a shock is referred to as left-handed), and with a minus sign if it is plotted clockwise (such a shock is referred to as right-handed). An oblique shock is characterized by a static pressure ratio $J = p_2/p_1$ (shock intensity). Subscript 1 corresponds to flow quantities before shock, and subscript 2 corresponds to flow quantities behind shock.

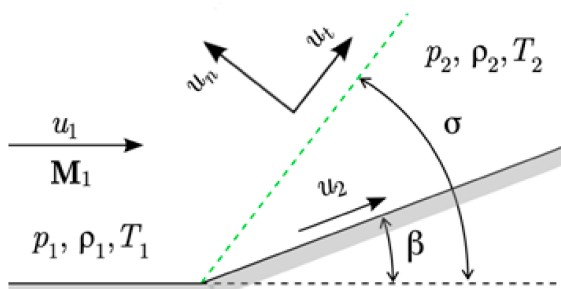

**Figure 2.** Oblique shock wave.

Oblique shocks are reflected from walls, intersect with other shock waves, and refract at tangential discontinuities. If there are many shock waves in the flow, they are usually numbered and indicated, with arrows gesturing to a given shock's direction; for example, $\vec{\sigma}_1$ or $\vec{\sigma}_5$. Left shocks are indicated by an arrow pointing to the left and right shocks are indicated by an arrow pointing to the right. The quantitiesbehind the corresponding shock are indicated by a number. For example, the quantities behind the shock are denoted by $M_1$ and $p_1$, and flow quantities behind the shock are denoted by $M_2$ and $p_2$.

Mass balance, momentum balance, and energy balance equations are written in the form:

Continuity equation

$$[\rho u_n] = \hat{\rho}\hat{u}_n - \rho u_n = 0. \tag{1}$$

Momentum equation in normal direction

$$\left[p + \rho u_n^2\right] = 0. \tag{2}$$

Momentum equation in tangential direction

$$[\rho u_n u_t] = 0. \tag{3}$$

Energy equation

$$[\rho u_n h_0] = 0. \tag{4}$$

Here, $u_n$ and $u_t$ are the projections of the velocity vector onto the directions that are normal in relation to the discontinuity plane and are tangential to it (Figure 1).

Equation (2) directly relates the change in pressure $p$ and dynamic pressure $\varrho u^2$ on a normal shock when the angle of inclination of the shock is $\sigma = 90°$. Using the Mach number M and the expression for the speed of sound $a^2 = \gamma p / \rho$ ($\gamma$ is the adiabatic index, $\gamma = c_p / c_v$, $c_p$ is the specific heat capacity at constant pressure, $c_v$ is the specific heat capacity at constant volume), and also by expanding the velocity vector of the incoming flow into the components $u_n$ and $u_t$, taking into account the fact that $u_{1t} = u_{2t}$, after simple transformations from (1) and (3), one can obtain an equation that relates the intensity of the shock with the angle of its inclination

$$J_\sigma = (1 + \varepsilon)\mathbf{M}^2 \sin^2 \sigma - \varepsilon. \tag{5}$$

The relation between angles $\beta$ and is $\sigma$ takes the form

$$\mathrm{tg}\beta = \frac{\mathbf{M}^2 \sin^2 \sigma - 1}{\frac{1}{1-\varepsilon}\mathbf{M}^2 - (\mathbf{M}^2 \sin^2 \sigma - 1)}\mathrm{ctg}\sigma. \tag{6}$$

Here, $\varepsilon = (\gamma - 1)/(\gamma + 1)$. This is the limit of the ratio of densities on the jump as $J \to \infty$.

Equations (5) and (6) are for a given **M** link, such as $J$, $\beta$, and $\sigma$. These equations refer to dynamic compatibility conditions on an oblique shock wave. For a given freestream Mach number, these equations define the closed curve (heart-shaped curve) shown in Figure 3. Since each M number has its own curve, shock polars are also called isomachs. It is convenient that the polar begins at the origin of coordinates [0,0]; therefore, it is usually constructed in the variables ($\ln J$, $\beta$).

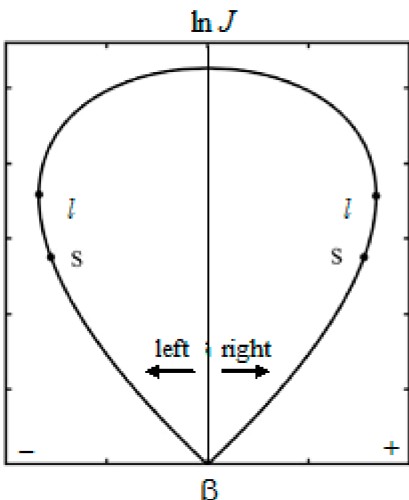

**Figure 3.** Shock polar (Busemann curve). Point $l$ corresponds to the shock with the maximum possible flow turning angle, and point $s$ is the sonic point.

## 3. Shock Wave Structure

The solution to the problem of calculating the interference of shock waves in one direction on the plane of Busemann's shock polars [26] is considered in [27,28]. In a shock wave structure formed during the interference of overtaking shocks, the reflected discontinuity $R$ can be either a compression shock or a rarefaction wave (Figure 1b). These two cases are separated by a characteristic shock wave structure, in which $R$ is a discontinuous characteristic, i.e., the intensity of $R$ is equal to one.

Figure 4 illustrates the case of interference of overtaking shocks with a reflected discontinuity (a shock wave). Figure 4a shows the determination of intensity ($\Lambda_4 = \ln J_4$) and the angle of flow ($\beta_4$) of the reflected discontinuity with the interference of overtaking shock waves, where $\beta_1$ and $\beta_2$ are the angles of flow at the wedges and M is the freestream Mach number. Figure 4b shows additional points of intersection of the third shock polar

that correspond to the reflected discontinuity *R*, with the first polar *A* and the second polar *B*, as well as point *C*, which corresponds to the solution with the reflected discontinuity (the shock wave).

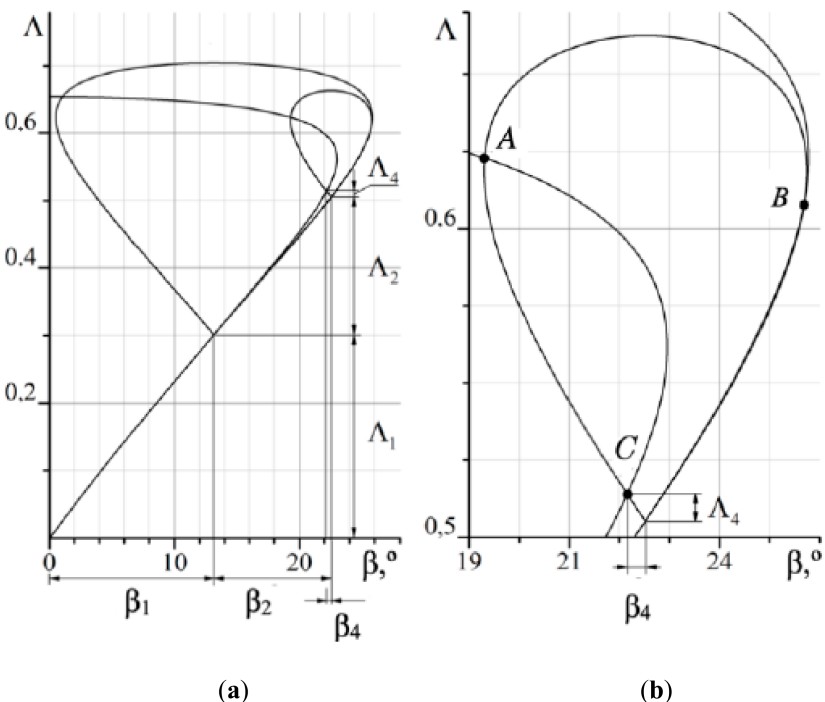

**(a)**                                        **(b)**

**Figure 4.** Analysis of the problem of analysis of overtaking shocks on the shock polar plane. (**a**) shows the determination of intensity ($\Lambda 4 = \ln J4$) and the angle of flow ($\beta 4$) of the reflected discontinuity with the interference of overtaking shock waves, where $\beta 1$ and $\beta 2$ are the angles of flow at the wedges and M is the freestream Mach number. (**b**) shows additional points of intersection of the third shock polar that correspond to the re-flected discontinuity R, with the first polar A and the second polar B, as well as point C, which corresponds to the solution with the reflected discontinuity (the shock wave).

In Figure 4, the first polar is plotted from Mach number M of the inlet flow. The second one is according to Mach number $M_1$ behind jump $\sigma_1$. The third one is according to Mach number $M_2$ behind jump $\sigma_2$. The solution corresponds to intersection point *C* of polar 1 and polar 3.

The intensity of reflected discontinuity $J_4$ is determined from the equations

$$\overleftarrow{\sigma}_4 - \beta_\sigma(M, J_4) + \beta_\sigma(M_2, GJ_4) = \beta_\sigma(M, J_1) + \beta_\sigma(M_1, J_2) \tag{7}$$

$$\overleftarrow{\omega}_4 - \beta_\sigma(M, J_4) - \beta_\omega(\hat{M}_2, GJ_4) = \beta_\sigma(M, J_1) + \beta_\sigma(\hat{M}_1, J_2) \tag{8}$$

where

$$G = (J_1 J_2)^{-1}; \; J_1 \in [1; J_s]; \; J_2 \in [1; J_{2s}] \tag{9}$$

Here, $\hat{M}_i$ is the Mach number behind the *i*th shock. $J_s$ and $J_{2S}$ are the intensities of the first and second shock, respectively, at which the flows behind them become sound, i.e., $M_{1,2} = 1$. Using the logarithms of the intensities $\Lambda = \ln J$ for the case of a reflected shock wave $R \equiv \sigma_4$, it is convenient to write the solution in the form of a system of two equations

$$\Lambda_1 + \Lambda_2 + \Lambda_4 = \Lambda_3; \; \beta_1 + \beta_2 - \beta_4 = \beta_3. \tag{10}$$

In system (10), $\beta$ angles are taken in absolute value without considering direction. Obviously, to satisfy conditions (10), the second shock polar must pass outside the main

one (Figure 4a). If it passes inside, the reflected discontinuity is a rarefaction wave $R \equiv \omega_4$. In some cases, the second polar may cross the first below the sound point. Such a polar configuration corresponds to the characteristic shock wave structure when discontinuity $R$ degenerates into a discontinuous characteristic. The second and third polars can also intersect with right-hand branches (point $B$ in Figure 4b), which leads to a more complex shock wave structure.

### 4. Domains of Existence of Regular Interference

The type of interference shown in Figure 1, when the flow behind all shocks is supersonic, is referred to as regular interference of overtaking shocks. This situation is not always possible. With a certain combination of parameters, a Mach stem is formed in the shock wave structure [29].

The flow behind the Mach stem is subsonic and total pressure drop $p_0$ is greater than on oblique shocks; therefore, this type of interference is undesirable in flow compression devices.

Sometimes the cause of Mach interference can be a counter shock. Point $A$ in Figure 4b corresponds to this case. However, this type of shock wave structure does not apply to the overtaking shocks.

Mach interference can also arise directly between shocks $\sigma_1$ and $\sigma_2$. As in the case of the reflection of an oblique shock wave from the wall, irregular (Mach) interference occurs when the third polar does not intersect with the first, and there is no solution corresponding to regular interference.

The boundary between two types of overtaking shock interference corresponds to the case when the third polar touches the first at one point (Figure 5). Figure 5 shows the boundaries of regular interference for different Mach numbers.

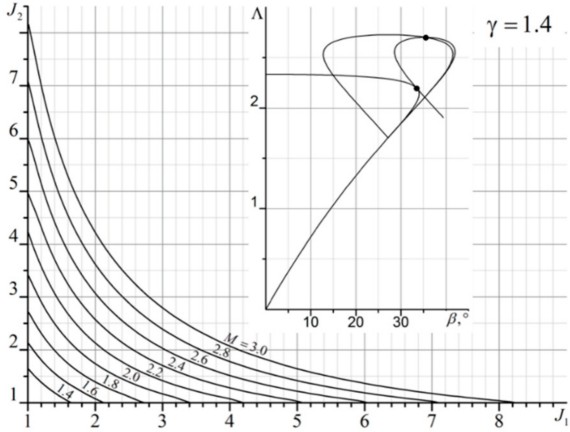

**Figure 5.** The boundary for solving the problem of regular interference.

### 5. Optimal Intake of Outer Compression

A typical external compression inlet has a wedge or cone with a variable angle; when flowing around it, several oblique shock waves of the same direction are formed (Figure 6). The system of oblique shocks is closed by a normal shock, which decelerates the flow to subsonic speed. Most often, there are two or three oblique shocks. If the number of corner points on the wedge increases, then it forms a curved surface.

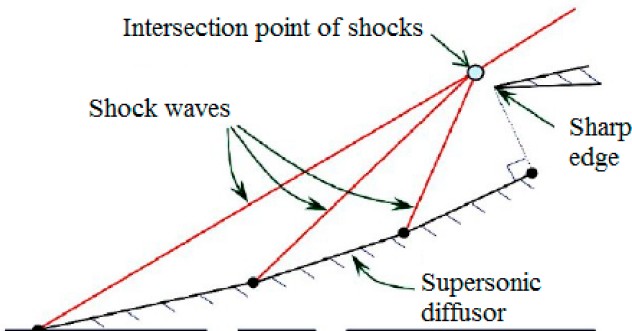

**Figure 6.** Shock wave structure consisting of shock waves in one direction in the supersonic air intake of external compression.

Assuming the shape of the surface is equal to the shape of the streamline in the centered Prandtl–Mayer compression wave, an isentropic air intake is designed in [10] (Figure 7). This air intake has the lowest possible total pressure loss of $p_0$.

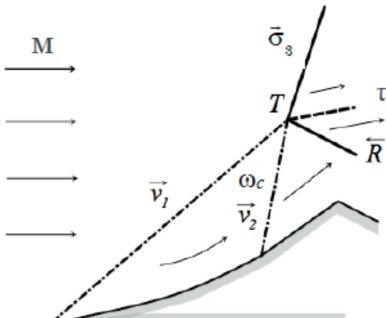

**Figure 7.** Centered compression wave $\omega_c$ as a limiting case of multiple discontinuities in one direction ($v_1$ and $v_2$ are discontinuous characteristics that limit the centered compression wave).

A criterion that must be satisfied by the shock wave structure, which is composed of overtaking shocks and a closing normal shock in order for the losses of $p_0$ to be minimal, is formulated in [10]. The normal components of the Mach number before all shocks must be the same. This condition follows directly from the formula for the total ratio of the total pressures behind a series of shocks

$$p_{0n}/p_0 = \prod_{i=1}^{n} J_i (E_i J_i)^{-(1+\varepsilon)/2\varepsilon} \tag{11}$$

where $\gamma$ is the ratio of specific heat capacities at constant pressure and constant volume, and $\varepsilon = (\gamma - 1)/(\gamma + 1)$, $E = (1 + \varepsilon J_i)/(J_i + \varepsilon)$.

Differentiating expression (11) and making result equal to zero, conditions for the equality of the ratio of the total pressures at any of the waves are derived. Hence, it follows that intensities $J_i$ of the oblique shocks should be equal, and the intensity of the closing normal shock should be slightly less; however, this small difference is usually neglected. This condition makes it possible to obtain the values of the inclination angles of the first shock ($\sigma$) and the second shock ($\sigma_2$) for any Mach number **M** > 1 (Figure 8). Figure 9 shows the total pressure values for the optimal air inlets with one and two oblique shocks.

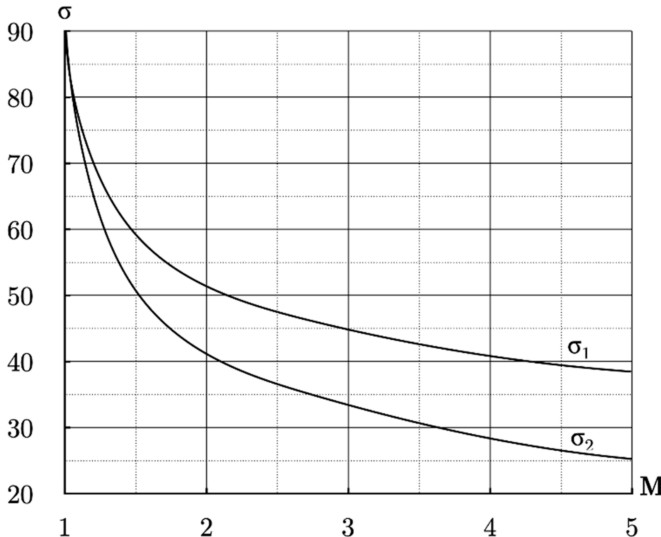

**Figure 8.** Optimal angles of inclination of overtaking shocks, ensuring minimum total pressure loss, $\gamma = 1.4$.

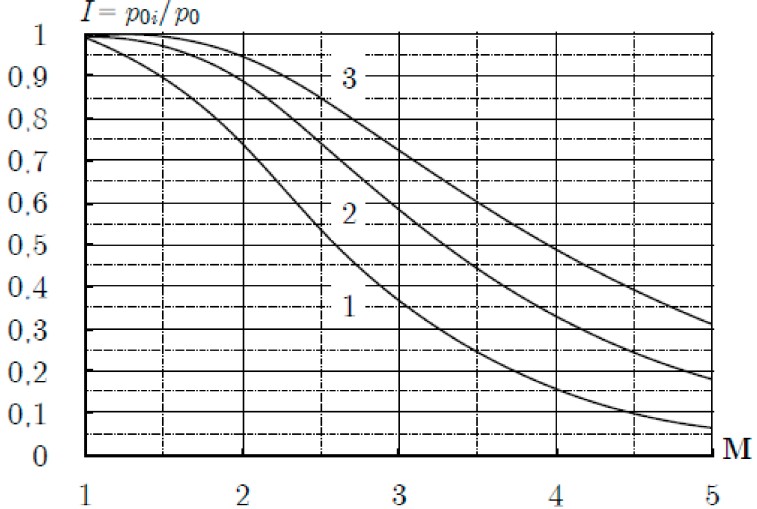

**Figure 9.** Dependence of the recovery factor of the total pressure behind the shock system on the Mach number, $\gamma = 1.4$. Line1 corresponds to the normal shock, line 2 corresponds to one oblique and one normal shock, and line 3 corresponds to two oblique shocks and one normal shock.

It can be seen that, in the target range from 2.6 to 3, the losses on the normal shock are almost two times higher than on the optimal shock wave structure, which is composed of two oblique shocks in the same direction and the closing normal shock.

## 6. Shock Wave Structure with Reflected Shock

Obviously, for the formation of shock wave structures in the air intake with a closing normal shock, reflected discontinuity $R$ must be a shock wave (Figure 1). Often, the angle of the wedge (cone) $\beta_1$ of the air intake must be specified for some design considerations. For example, the maximum diameter of the cone is limited by the midsection of an aircraft's fuselage, and its length is limited by the dimensions of the equipment that is installed inside.

Then, it is possible to formulate the problem to determine the types of reflected discontinuity in the shock wave structure, which are formed as the flight Mach number and angle $\beta_1$ vary. Figures 10 and 11 show the domains of existence of the corresponding shock wave structure.

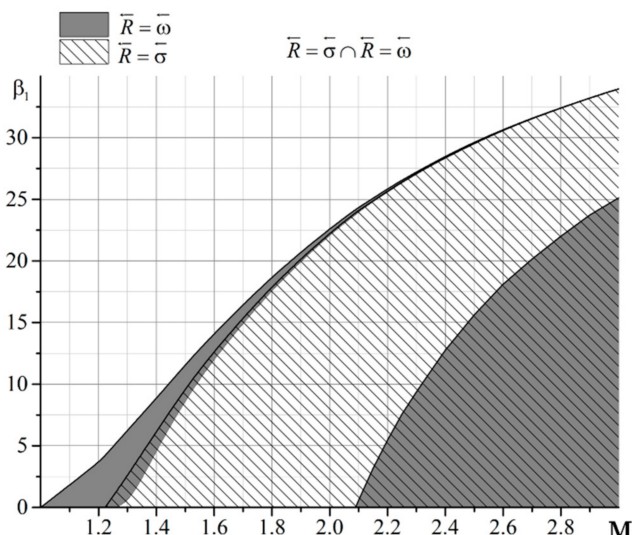

**Figure 10.** Domains of existence of different types of reflected discontinuity depending on Mach number M and the flow turn angle at the first shock, $\beta_1$, $\gamma = 1.4$.

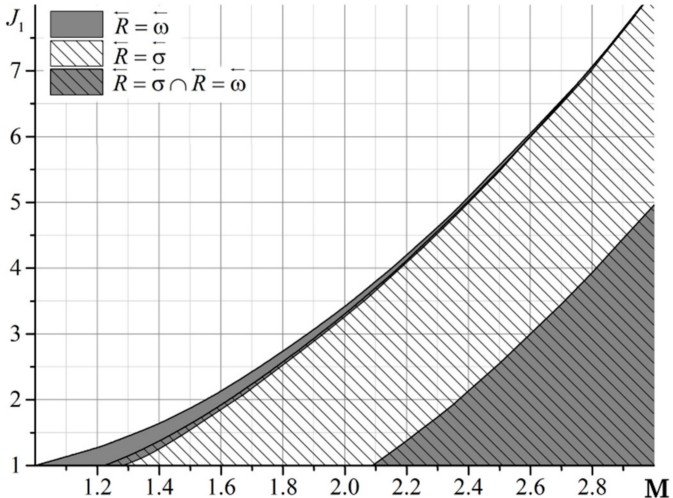

**Figure 11.** Domains of existence of different types of reflected discontinuity depending on Mach number M and the intensity of the first shock, $J_1$, $\gamma = 1.4$.

As can be seen in Figures 10 and 11, the domain in which a shock wave structure with a reflected discontinuity exists is relatively narrow. There are many more areas in which the existence of a reflected rarefaction wave is possible, as well as areas where the reflected discontinuity can be either a rarefaction wave or a shock wave. Consequently, the regions of existence of shock wave structures with different types of reflected discontinuity should be studied on the $J_1 - J_2$ plane. The cases $R \equiv \sigma$ and $R \equiv \omega$ are separated by the characteristic shock wave structure, in which

$$J_1(\mathbf{M}) = J_2(\mathbf{M}_1), \quad \beta_1(\mathbf{M}) = \beta_2(\mathbf{M}_1). \tag{12}$$

Equation (12) allows us to construct the characteristic curve $J_1(J_2)$ for any freestream Mach number (Figure 12). In Figure 12, the dotted line separates the domains in which $R \equiv \sigma$ lies above or below the line corresponding to the given Mach number.

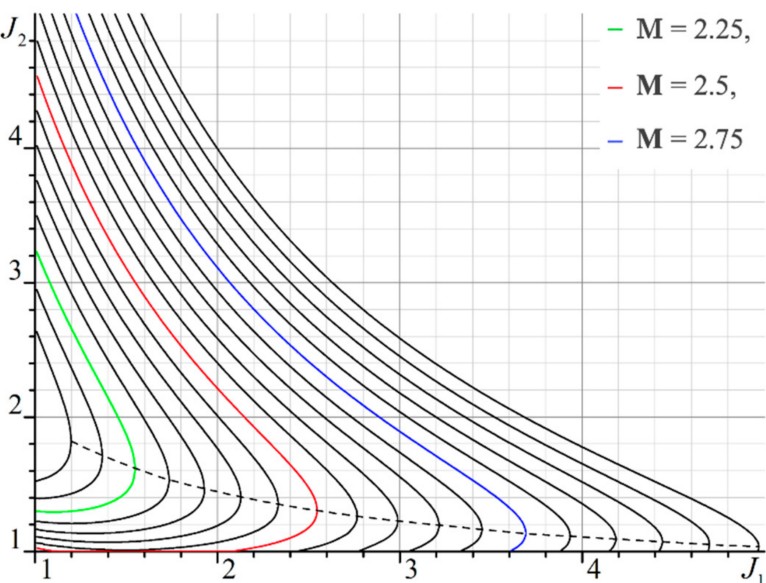

**Figure 12.** Lines $J_1$–$J_2$, corresponding to the characteristic shock wave structure in the range of Mach numbers **M** = 2.15–3, $\gamma$ = 1.4.

In Figure 13, for M = 3, the domains of existence of the shock wave structure with different types of reflected discontinuity are shown. The dotted line corresponds to the sound intensity of the second shock, above which regular interference is impossible. As seen in Figure 14, there is one more branch of a characteristic shock wave structure for high values $J_1$ (not shown in Figure 13); however, it has no practical importance.

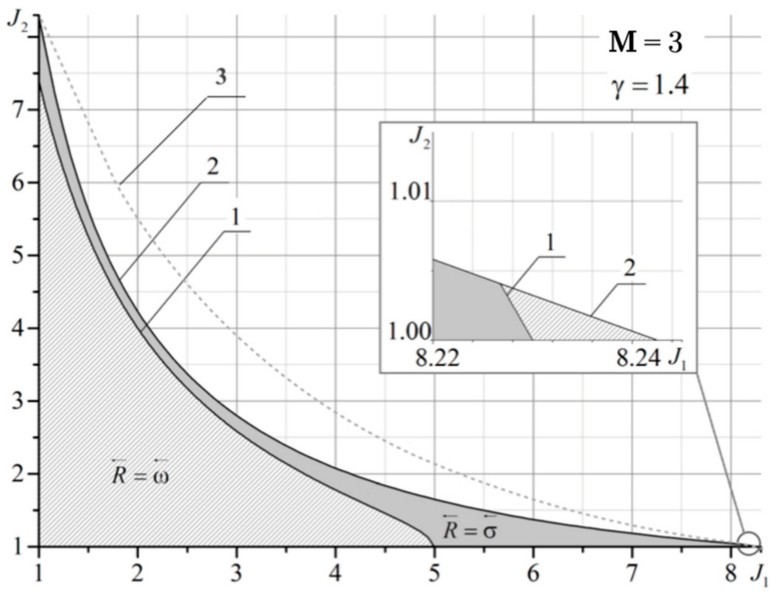

**Figure 13.** Domains of existence of various types of reflected discontinuity with regular interference of shock waves in the same direction. Line 1 corresponds to characteristic shock wave, line 2 corresponds to boundary of regular interference, and line 3 corresponds to sound intensity $J_s$ ($M_2$ = 1) of the second shock.

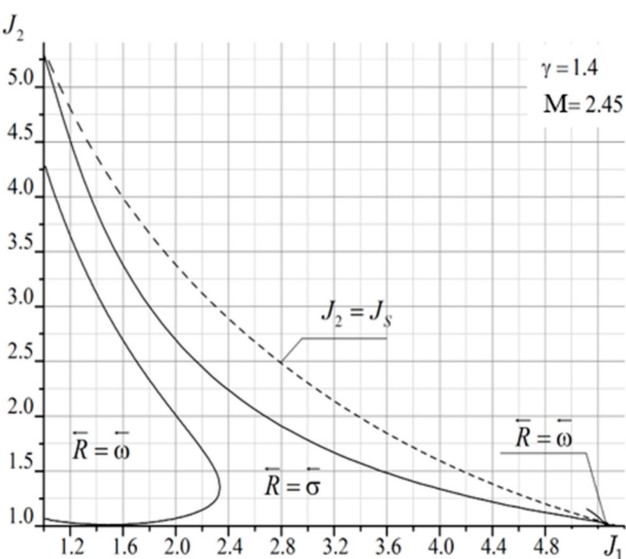

**Figure 14.** Domains of existence of various types of reflected discontinuity.

As the Mach numbers decrease, region $R \equiv \omega$ shifts to the left and upward, until, at M = 2.45, another region $R \equiv \sigma$ appears in the region of small $J_1$ (Figure 14).

This moment corresponds to the tangency of the iso-Mach line contour at M = 2.45 to the horizontal axis in Figure 12. With a further decrease in the Mach number, the iso-Mach line contours are located entirely above the horizontal axis (Figure 14). To design the optimal external compression inlet with two oblique shocks, one can choose any point in Figures 13–15 belonging to region $R \equiv \sigma$, so that $J_1 = J_2$. In the limiting case, it can lie on the iso-Mach line contours shown in Figure 10. It can be seen that, for Mach numbers close to M = 3, region $R \equiv \sigma$, in which two shocks can be chosen so that condition $J_1 = J_2$ is satisfied, is very narrow.

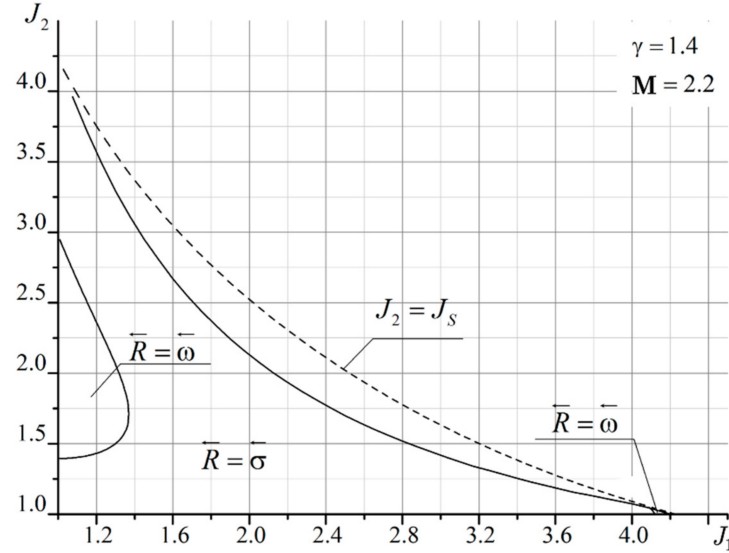

**Figure 15.** Domains of existence of various types of reflected discontinuity for **M** = 2.2.

Thus, at high Mach numbers, the design of an optimal air intake with two oblique shocks is difficult, and it is necessary to examine other schemes. Figures 16–18 show the values of the intensity of the reflected discontinuities.

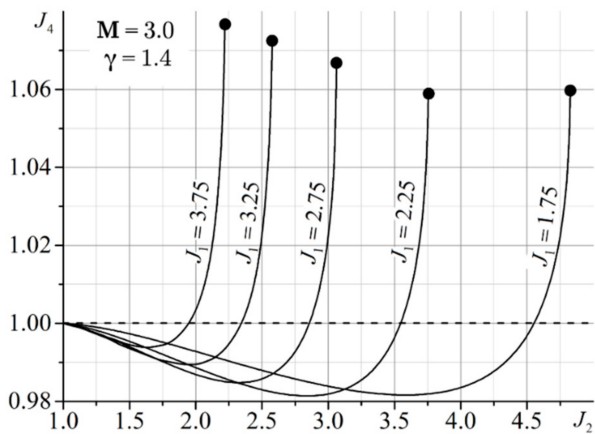

**Figure 16.** Dependence of the intensity of reflected discontinuity $J_4$ on the intensity of the first $J_1$ and second $J_2$ overtaking shocks for **M** = 3.

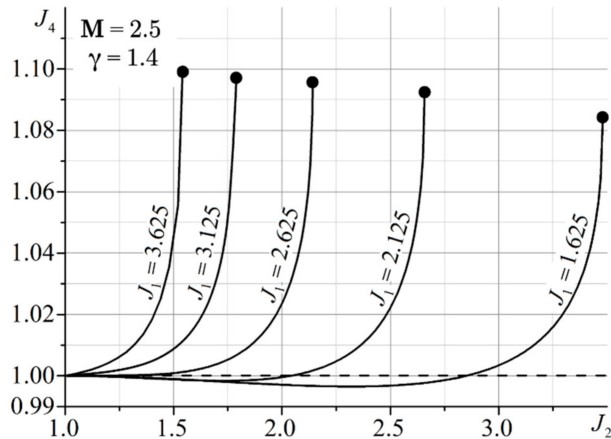

**Figure 17.** Dependence of the intensity of reflected discontinuity $J_4$ on the intensity of the first $J_1$ and second $J_2$ overtaking shocks for **M** = 2.5.

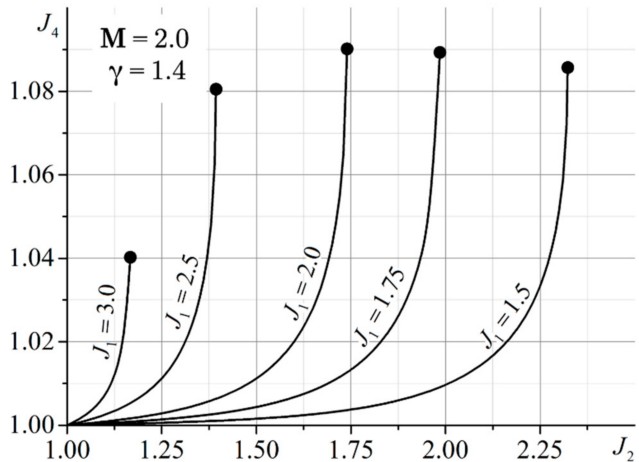

**Figure 18.** Dependence of the intensity of reflected discontinuity $J_4$ on the intensity of the first $J_1$ and second $J_2$ overtaking shocks for **M** = 2.

It can be seen that, in the most unfavorable case, the intensity of the rarefaction wave does not exceed 1% of the total compression ratio of the flow in all shock waves.

## 7. Conclusions

The interference of shock waves that deflect the flow in one direction (overtaking shock waves) is considered. For different Mach numbers, the boundaries of regular interference are plotted, at least when the flow at all points behind the shocks remains supersonic. Information on optimal shock wave structures with overtaking shock waves is provided in this study. We have shown that the intensity of all oblique shock waves in such structures should be the same. The domains of existence of shock wave structures, in which the reflected discontinuity is formed during the interference of overtaking shock waves, is a compression shock or a rarefaction wave. Iso-Mach line contours, which determine the characteristic shock wave structure that separates the two cases for different Mach numbers, are also plotted. Although the reflected rarefaction wave is undesirable in compression devices, its intensity is about 1% of the total compression ratio of all oblique shock waves.

The results obtained in the study can potentially be useful in the design of supersonic intakes and advanced jet engines. The domains of existence of regular interference of overtaking shocks are also provided in this study. These results can be used in the design of air intakes, for which irregular interference is an off-design and often an emergency mode. On the other hand, the domains of existence of reflected discontinuities (shock waves) are constructed, and their intensity is determined. Shock wave structures with a reflected discontinuity are structurally stable in air intakes, i.e., they do not fall apart with a small disturbance in incident flow quantities.

**Author Contributions:** Conceptualization, P.B. and I.V.; methodology, P.B.; software, I.V.; validation, K.V.; formal analysis, P.B.; investigation, I.V.; resources, K.V.; writing—original draft preparation, P.B.; writing—review and editing, K.V.; visualization, I.V.; supervision, P.B. All authors have read and agreed to the published version of the manuscript.

**Funding:** This work was financially supported by the Ministry of Science and Higher Education of Russian Federation during the implementation of the project "Creating a leading scientific and technical reserve in the development of advanced technologies for small gas turbine, rocket and combined engines of ultra-light launch vehicles, small spacecraft and unmanned aerial vehicles that provide priority positions for Russian companies in emerging global markets of the future", No. FZWF-2020-0015.

**Conflicts of Interest:** The authors declare no conflict of interest.

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
