# Peer review of "Interaction between Shock Waves Travelling in the Same Direction"

_fluids, doi:10.3390/fluids6090315_

Round 1

Reviewer 1 Report

The paper examines, analytically, the interaction of oblique shocks in the steady state. The limits of the regular reflections are computed. Irregular reflections are tangentially studied.

I think this paper deserves publication in Fluids. However, I suggest some minor and moderate changes to be considered by the authors.

First, the literature review is poor. Well-known researchers on the topic are missed. For example, Henderson, Ben-Dor, Abd-el-Fattah, Landau, Hornung,... are some classic authors with a known record on the topic. See also Sternberg, Joseph. "Triple‐Shock‐Wave Intersections." The Physics of Fluids 2.2 (1959): 179-206. Consideration of the mentioned papers (studied and commented) will enrich the analysis of the present work. This is a must in a theoretical work like the one proposed in this work.

Second, irregular reflections are superficially addressed. Recent studies like the theoretico-numerical work by Martínez-Ruiz, Daniel, et al. "Irregular self-similar configurations of shock-wave impingement on shear layers." Journal of Fluid Mechanics 872 (2019): 889-927 can be useful to complete the analysis. That is, the nature of the irregular reflection and their dynamics can be easily anticipated. This will help to provide limits of validity over downstream influences.

Third, the journal audience is expected to be the fluid mechanics community. The mathematical formulation should be written from the fluid mechanics foundations. Oblique-shock Rankine-Hugoniot equations must be provided in terms of arothermal properties. 

Fourth, the order of the system of equations must be analyzed along with the number of possible solutions. That is, conservation equations by themselves cannot be used to disregard spurious solutions from the multiplicity of solutions that is expected to appear in high-order systems. In absence of numerical simulations or laboratory experiments, entropy generation and boundary conditions influences must be considered to get the justified final solutions.

Reviewer 2 Report

Comments on Manuscript ID fluids-1284934
Title: Interaction of shock waves travelling in the same direction

The motivation must be clarified. The introduction must be rewritten to demonstrate the gap in literature. The references are outdated and only 1 recent reference has been cited.  
As the author proposed the results of the study can have application in design of supersonic intakes jet engines. It is suggested that based on the obtained results, the authors illustrate the Implementation of these result.

Round 2

Reviewer 2 Report

the revisions have been addressed and the revised version is satisfactory